# Validity of Prediction Equations of Maximal Heart Rate in Physically Active Female Adolescents and the Role of Maturation

**DOI:** 10.3390/medicina55110735

**Published:** 2019-11-13

**Authors:** Sophia D. Papadopoulou, Sousana K. Papadopoulou, Foteini Alipasali, Dimitris Hatzimanouil, Thomas Rosemann, Beat Knechtle, Pantelis T. Nikolaidis

**Affiliations:** 1Laboratory of Evaluation of Human Biological Performance, Department of Physical Education & Sport Science, Aristotle University of Thessaloniki, 57001 Thessaloniki, Greece; sophpapa@phed.auth.gr; 2Department of Nutritional Sciences and Dietetics, International Hellenic University, 57400 Thessaloniki, Greece; sousana@nutr.teithe.gr; 3Department of Physical Education & Sport Science, Aristotle University of Thessaloniki, 62100 Serres, Greece; alip_fotini@yahoo.gr; 4Laboratory of Evaluation of Human Biological Performance, Department of Physical Education & Sport Science, Aristotle University of Thessaloniki, 57001 Thessaloniki, Greece; xatjiman@phed.auth.gr; 5Institute of Primary Care, University of Zurich, 8091 Zurich, Switzerland; thomas.rosemann@usz.ch; 6Exercise Physiology Laboratory, 18450 Nikaia, Greece; pademil@hotmail.com

**Keywords:** cardiac rate, exercise prescription, exercise testing, prediction equations, training zones, volleyball

## Abstract

*Background and objectives*: Maximal heart rate (HR_max_) is an important training and testing tool, especially in the context of evaluating intensity in exercise prescription; however, few studies have examined the validity of prediction equations of HR_max_ in physically active female adolescents and the role of maturation level. Therefore, the aim of the present study was to examine the differences between measured and predicted HR_max_ in a sample of physically active female adolescents. *Materials and Methods*: Seventy-one selected volleyball players (age 13.3 ± 0.7 years, body mass 62.0 ± 7.2 kg, height 1.72 ± 0.06 m) performed a 20 m shuttle run endurance test, and the actual HR_max_ was compared with Tanaka HR_max_ (‘208 − 0.7 × age’) and Fox HR_max_ (‘220 − age’). *Results*: A large main effect of assessment method on HR_max_ was found (*p* < 0.001, *η*^2^ = 0.486) with Fox overestimating actual HR_max_ by 6.8 bpm (95% confidence intervals, CI; 4.2, 9.3) and Tanaka underestimating actual HR_max_ by −2.6 bpm (95% CI; −5.1, −0.1). The more matured participants had similar actual HR_max_ (mean difference −2.4 bpm; 95% CI; −6.5, 1.7; *p* = 0.242, d = −0.28), difference Fox − actual HR_max_ (1.5 bpm; 95% CI; −2.6, 5.6, *p* = 0.466, d = 0.17), and difference Tanaka − actual HR_max_ (1.7 bpm; 95% CI; −2.4, 5.8; *p* = 0.414, d = 0.19) to the less matured participants. *Conclusions*: These findings suggest that age-based prediction equations of HR_max_ developed in adult populations should be applied with caution in physically active female adolescents, and Tanaka should be preferred instead of the Fox equation.

## 1. Introduction

Exercise prescription in a health or sport context includes information about the basic characteristics of exercise, e.g., mode, duration, sets, repetitions and intensity [1]. An accurate prescription of exercise intensity is crucial to induce optimal chronic adaptations to exercise. In aerobic exercise, exercise intensity should be within a specific range (‘training zone’) to elicit desired physiological adaptations. Exercise intensity, when evaluated by heart rate (HR) measures, is usually expressed as percentage of the maximal HR (HR_max_) [2]. Thus, the knowledge of HR_max_ is essential to accurately prescribe exercise intensity.

HR_max_ might be measured directly through a graded exercise test (GXT) till exhaustion, whereas indirectly it might be calculated using an age-based equation [3]. Occasionally, a practitioner might wish to use an age-based equation when a GXT till exhaustion would be contraindicated (e.g., exercise testing of patients or fitness assessment of athletes during a competitive period). In such cases, popular age-based equations—e.g., Fox, Naughton and Haskell (Fox HR_max_; ‘220 − age’) [4] and Tanaka, Monahan and Seals (Tanaka HR_max_; ‘208 − 0.7 × age’) [5] —might be used. The validity of these equations has been examined in both adults [6] and adolescents [7,8]. In adolescents, the need for further studies to control for the potential effect of biological age on sympathetic modulation of HRmax was identified [8].

Although the existing research on the validity of age-based predicted HR_max_ has enhanced our understanding of the practical applications of these equations, the role of maturation as a covariate has not been considered adequately so far by the existing literature, where two relevant studies were found [9,10]. However, pubertal status was categorized using only a chronological cut-off age (14 years) for females in one study [9], whereas a mixed sample of females and males was considered in the other study, which included maturation offset in regression analysis instead of comparing maturation groups [10]. Since it has been suggested that age-based prediction equations developed in adults were not applicable in children [8], it would be reasonable to assume that maturation would be expected to be related to the validity of such equations. Therefore, the aim of the present study was to examine the validity of two popular prediction equations (Fox and Tanaka) in female adolescent volleyball players and the variation of validity by maturation level.

## 2. Materials and Methods

For the purpose of the study, a sample of convenience consisting of 71 selected female adolescent volleyball players was retrospectively analyzed. Participants were members of volleyball clubs from Attiki, i.e., the wider geographical area of Athens, and were selected by the national teams’ coaches to be considered as members of national teams. Inclusion criteria were the absence of any known pain or injury that would prevent participants from exercise testing. Parents or guardians of participants provided informed consent prior to exercise testing. The study was approved by the local institutional review board (Exercise Physiology Laboratory, Nikaia, Greece; number EPL2008/1, 6 August 2008). All procedures adhered to the 2013 revision of the Declaration of Helsinki.

Exercise testing was performed in two sessions separated by a week. Anthropometric characteristics were recorded in the first session in an exercise physiology laboratory, and a 20 m shuttle run test (SRT) [11] was done in the second session in an indoor court. With regards to anthropometric measures, weight and height were examined using an electronic weight scale (HD-351, Tanita, Arlington Heights, IL, USA) and a portable stadiometer (SECA, Leicester, UK), respectively. A calliper (Harpenden, Burgess Hill, UK) for skinfold thickness (0.5 mm) measured skinfold thickness of 10 sites (cheek, wattle, chest I, chest II, triceps, subscapular, abdominal, suprailiac, thigh and calf), and body fat percentage was calculated from the sum of these skinfolds’ thicknesses [12]. Chronological age was estimated by a table of decimals of year [13]. Peak height velocity (PHV) was used to assess biological maturation, and age at PHV (APHV) was predicted based on sex, date of birth, date of measurement, height, sitting height and body mass [14]. Thereafter, difference (ΔAPHV) between chronological age and APHV was considered as an estimate of biological age. The estimation of biological maturation using the approach of Mirwald and colleagues relied on the differential timings of growth of height, sitting height and leg length [14]. For the purpose of the present study, biological maturation was expressed using a continuous variable to allow regression analysis and comparison to chronological age. Actual HR_max_ was recorded as the peak HR during SRT. SRT started at 8.5 km/h, and speed increased by 0.5 km/h every minute till exhaustion [11]. During this test, HR was monitored by Team2 (Polar Electro Oy, Kempele, Finland). In addition, HR_max_ was predicted using Fox HR_max_ (‘220 − age’) [4] and Tanaka age-based equations (‘208 − 0.7 × age’) [5].

Statistical analyses were performed using GraphPad Prism v. 7.0 (GraphPad Software, San Diego, CA, USA) and IBM SPSS v. 23.0 (SPSS, Chicago, IL, USA). Significance was set at *p* = 0.05. Data were tested for normality using the Kolmogorov-Smirnov test and visual inspection of Q-Q plots and were expressed as mean and standard deviations of the mean (SD). A one-way repeated measures analysis of variance (ANOVA) examined differences between actual, Fox and Tanaka HR_max_. Despite potential limitations (e.g., unequal variances), this statistical approach was used previously by studies on differences between actual and age-based predicted equations [3,10,15] and was selected for the present study to provide comparable findings. The magnitude of differences in ANOVA was evaluated using eta squared, classified as small (0.01 < *ŋ*^2^ ≤ 0.06), medium (0.06 < *ŋ*^2^ ≤ 0.14) and large (*ŋ*^2^ > 0.14) [16]. The accuracy and variability of Fox and Tanaka HR_max_ were examined using Bland–Altman analysis. The relationship of actual HR_max_ and age was tested by Pearson’s product–moment correlation coefficient (*r*). The magnitude of *r* was interpreted as very small (*r <* 0.1), small (0.1 *≤ r* < 0.3), moderate (0.3 *≤ r* < 0.5), large (0.5 *≤ r* < 0.7), very large (0.7 *≤ r* < 0.9), nearly perfect (0.9 *≤ r* < 1) and perfect (*r* = 1) [17]. A *t*-test examined differences between less (n = 37, ΔAPHV ≤ 1.8 years) and more matured participants (n = 34; ΔAPHV ≥ 1.9 years). The classification of participants into these two groups relied on the split technique, which—without neglecting its drawbacks—has been a common practice in clinical research [18]. The magnitude of differences in *t*-test was evaluated using Cohen’s *d*, which was considered as trivial (*d* ≤ 0.2), small (0.2 < *d* ≤ 0.6), moderate (0.6 < *d* ≤ 1.2), large (1.2 < *d* ≤ 2.0) or very large (*d* > 2.0) [17]. Complimentary analyses included comparison of squared differences (predicted - actual HR_max_) by maturation level (Appendix A), regression analysis of these squared differences with maturation (Appendix A) and mixed model analysis (Appendix A). 

## 3. Results

The demographic data of participants by maturation group are presented in Table 1. The more matured participants were older, heavier and taller than their less matured counterparts (*p* < 0.05), whereas no difference in body mass index (BMI) and SRT between the two maturation groups was observed. The actual and predicted HR_max_ can be seen in Table 2. No difference in actual HR_max_ was shown between the maturation groups (*p* > 0.05). The more matured participants had similar actual HR_max_ (mean difference −2.4 bpm; 95% confidence intervals, CI; −6.5, 1.7; *p* = 0.242, d = −0.28), difference Fox − actual HR_max_ (1.5 bpm; 95% CI; −2.6, 5.6, *p* = 0.466, d = 0.17) and difference Tanaka − actual HR_max_ (1.7 bpm; 95% CI; −2.4, 5.8’ *p* = 0.414, d = 0.19) to the less matured participants. No relationship was observed of actual HR_max_ with chronological age or ΔAPHV (Figure 1).

ΔAPHV = difference from the age at peak height velocity; error bars represent 95% confidence intervals.(1)

A large main effect of assessment method on HR_max_ was found (*p* < 0.001, *η*^2^ = 0.486), with Fox overestimating by 6.8 bpm (95% CI; 4.2, 9.3) and Tanaka underestimating actual HR_max_ by −2.6 bpm (95% CI; −5.1, −0.1) (Figure 2). The Bland–Altman plots (Figure 3) show that overall, Fox overestimated and Tanaka underestimated actual HR_max_; however, a similar trend was observed in both cases, where there was an overestimation when actual HR_max_ was low and an underestimation when actual HR_max_ was high.

## 4. Discussion

The present study examined the application of two popular age-based prediction equations of HR_max_ in physically active female adolescents. The main findings were that (a) Fox’s ‘220 − age’ overestimated actual HR_max_ by ~7 bpm, (b) Tanaka’s ‘208 − 0.7 × age’ underestimated actual HR_max_ by ~3 bpm, (c) a large amount of individual variation was observed in both prediction equations, and (d) no variation of these findings was observed by maturation. Tanaka underestimated HR_max_ less than Fox overestimated, and the notion that the former provided a better estimate of actual HR_max_ than Fox was in agreement with the existing literature in adolescents [8]. Particularly, a meta-analysis of seven articles reported an underestimation using Tanaka by ~3 bpm and an overestimation using Fox by ~12 bpm [8]. In addition, the large amount of individual variation was in agreement with previous studies on adolescents [10], and this variation was larger in studies using wider ranges of ages [8]. An explanation of this variation might be that HR_max_ was related to several factors, such as HR at rest (the higher the HR at rest, the higher the HR_max_) [9] and body mass (the higher the body mass index, the lower the HR_max_) [19].

The lack of correlation of actual HR_max_ with chronological or biological age was not surprising and should be attributed to the small range (<3 years) of chronological age of participants (11.7–14.6 years). Previous studies [6,15,20], which observed negative correlation of large to very large magnitude between actual HR_max_ and chronological age, covered a chronological range of many decades. Furthermore, growth-related differences in HR might be linked to maturation of sympathetic-parasympathetic neural regulation and the role of circulating modulators [21,22]. In addition, it was assumed that the older participants would exhibit lower HR_max_ than their younger counterparts due to their larger accumulated training experience. This assumption relied on the suggestion that aerobic training would result in chronic adaptations, such as plasma volume expansion and alteration of the electrophysiology of the sinoatrial node [23]. A meta-analysis reported that aerobic training in sedentary individuals and athletes decreased HR_max_ by 6 bpm, and detraining and tapering increased HR_max_ by 6 bpm, whereas athletes had lower HR_max_ than sedentary individuals by 8 bpm [23]. The lack of differences in actual HR_max_ by age might also be attributed to the nature of volleyball training practiced by participants, which should not be considered aerobic training.

With regards to the role of maturation, no difference in the validity of age-based prediction equations was observed between the two maturation groups. The research hypothesis was that the more matured group would have lower actual HR_max_, considering that an age-related decline in HR_max_ was more striking after puberty with a rate of ~0.7 bpm per year [24]. Prepubescents would be expected to show blunted sympathetic modulation during exercise compared to post-pubescents and adults due to age-related differences in sympathoadrenal regulation [25]. The lack of differences observed between maturation groups observed in the present study might be attributed to the small range of ΔAPHV (~2 years, Figure 1b).

A limitation of this study was that actual HR_max_ was elicited during a field GXT. Thus, caution would be needed to generalize the findings in a laboratory setting. It was acknowledged that laboratory criteria of achieving maximal performance (e.g., plateau in oxygen uptake, carbon dioxide to oxygen exchange ratio, lactate) [26] were not applied. On the other hand, all participants were motivated to perform maximally, as the testing session occurred in the context of selection process for the national teams. In addition, a field GXT might result in even higher HR_max_ than a laboratory GXT on treadmill ergometer, likely due to the more natural running patterns observed in the former than in the latter case [27]. A strength of the study was its novelty, as it was the first one—to the best of our knowledge—to examine the role of maturation in predicting HR_max_ in physically active adolescents. Considering the importance of exercise intensity when prescribing exercise programs [1], the findings of the present study would have practical applications for practitioners in the context of testing and training. Since Fox overestimated actual HR_max_, its use should be contraindicated when there is high exercise intensity and, consequently, increased risk of overtraining, whereas Tanaka should be avoided in prescribing low exercise intensity, since an underestimation of actual HR_max_ would result in a training program of inadequate exercise intensity to induce the desired physiological adaptations. Another limitation of the findings of the present study was the sample size, which was relatively small due to the specific characteristics of participants (i.e., selected athletes). Future studies should not only use larger sample size, but also recruit participants with a larger range of maturity status, e.g., from pre-pubescence to post-pubescence.

## 5. Conclusions

Based on these findings, it is suggested that age-based prediction equations of HR_max_ developed in adult populations should be applied with caution in physically active female adolescents. In the context of testing and training, practitioners would be advised to choose an age-based prediction equation considering the risks of overestimation and underestimation of actual HR_max_. Tanaka equation provided more accurate values than Fox equation.

## Figures and Tables

**Figure 1 medicina-55-00735-f001:**
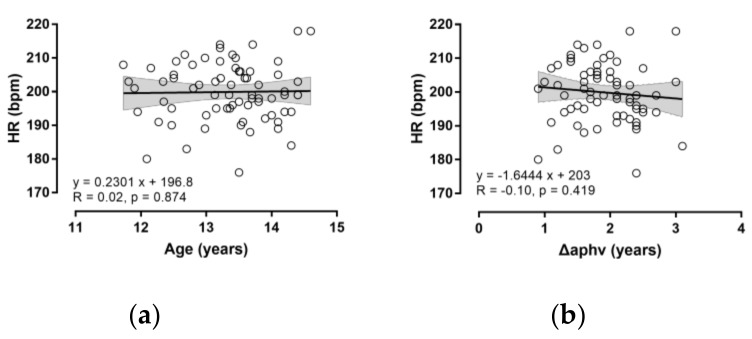
Relationship of actual HR_max_ with chronological age (**a**) and difference from the age at peak height velocity (**b**).

**Figure 2 medicina-55-00735-f002:**
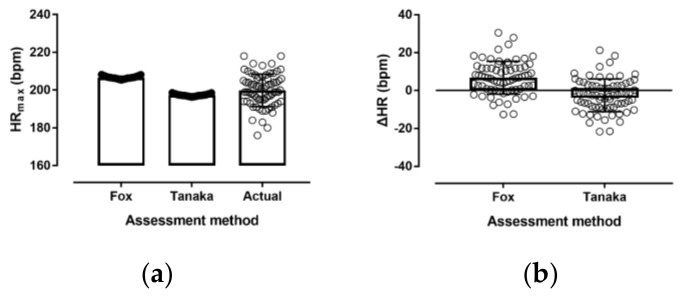
Variation of HR_max_ by assessment method (**a**) and difference (ΔHR) of Fox and Tanaka values compared to actual HR_max_ (**b**). Individual values are presented by open circles (○), boxes show means, and error bars refer to standard deviations.

**Figure 3 medicina-55-00735-f003:**
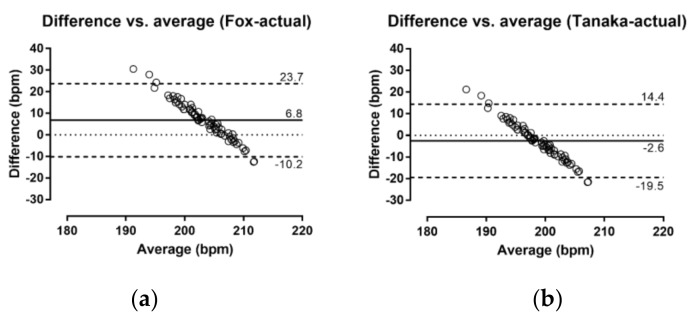
Bland–Altman plots of the difference of Fox (**a**) and Tanaka (**b**) with actual HR_max_. Solid lines represent agreement, whereas the upper and lower dashed lines show 95% limits of agreement. The values of agreement and upper and lower 95% limits of agreement are presented within the figure. ‘Difference’ refers to the difference between predicted and actual HR_max_, and ‘average’ was calculated using the formula (predicted HR_max_ + actual HR_max_)/2.

**Table 1 medicina-55-00735-t001:** Demographic characteristics of participants.

Variable	Total (n = 71)	Less Matured (n = 37)	More Matured (n = 34)
Age (years)	13.3 ± 0.7	12.9 ± 0.7	13.8 ± 0.4 *
ΔAPHV (years)	1.9 ± 0.5	1.5 ± 0.3	2.3 ± 0.3 *
Weight (kg)	62.0 ± 7.2	59.1 ± 6.1	65.1 ± 7.2 *
Height (m)	1.72 ± 0.06	1.68 ± 0.05	1.75 ± 0.04 *
BMI (kg.m^-2^)	21.1 ± 2.2	20.9 ± 2.1	21.2 ± 2.4
BF (%)	21.2 ± 4.6	20.8 ± 4.6	21.6 ± 4.5
SRT (min:s)	5:00 ± 1:17	5:08 ± 1:20	4:51 ± 1:13

BMI = body mass index; BF = body fat percentage; SRT = 20 m shuttle run test; * *p* < 0.05.

**Table 2 medicina-55-00735-t002:** Actual, Fox and Tanaka HR_max_ of participants.

Variable	Total (n = 71)	Less Matured (n = 37)	More Matured (n = 34)
Actual HR_max_ (bpm)	199.9 ± 8.6	201.1 ± 8.4	198.7 ± 8.8
Fox HR_max_ (bpm)	206.7 ± 0.7	207.1 ± 0.7	206.2 ± 0.4 *
Tanaka HR_max_ (bpm)	197.3 ± 0.6	197.7 ± 0.5	197.0 ± 0.3 *
Fox - Actual HR_max_ (bpm)	6.8 ± 8.7	6.0 ± 8.5	7.6 ± 8.9
Tanaka - Actual HR_max_ (bpm)	−2.6 ± 8.6	−3.4 ± 8.5	−1.7 ± 8.8

HR_max_ = maximal heart rate; * *p* < 0.05.

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
