# Peer review of "Validity of Prediction Equations of Maximal Heart Rate in Physically Active Female Adolescents and the Role of Maturation"

_1010-660X, 2019, doi:10.3390/medicina55110735_

Round 1

Reviewer 1 Report

In their manuscript ‚Validity of prediction equations of maximal heart rate in physically active female adolescents’, Papadopoulou et al. investigate the validity of two age-based formulas that have the purpose to predict an individual’s maximal heart rate. One might argue that cases where maximal heart rate in young individuals is not determinable and rather has to be estimated using a formula for training purposes are quite exceptional. Nevertheless, it is important to investigate to what extent these formulas are valid in young individuals. Thus, the research question indeed has scientific merit.

The author’s conclusion is that “age-based prediction equations of HRmax developed in adult populations should be applied with caution in physically active female adolescents”. I agree to this conclusion, however, without knowledge of their data I would also have recommended caution in this regard. However, my impression is that the comparison of the two formulas represents the most relevant aspect of this study. This comparison is not statistically analysed in a sufficient manner. Thus, I would recommend publication of this manuscript in ‘Medicina’, given that some major revisions are made.

My specific points are the following:

It is unclear how the sample size was determined. I assume that the institutional review board required a sample size calculation in 2008. Please provide a copy of the original study protocol, ideally with proof that this version of the study protocol was the approved one (e.g. with a QR-code, stamp). From the study protocol it should also be clear whether the primary purpose of the study was to compare the formulas in a prospective study, or whether this was actually retrospective data analysis. In this case, the retrospective nature should clearly be highlighted in the manuscript together with the original purpose of the study. Were data collected solely for the purpose of comparing the two formulas? 2: It is valuable that normal distribution was tested, however, there are several other assumptions of one-factorial repeated measures ANOVA, e.g. sphericity. Given the huge variance of the actual HRmax data in Fig. 2 compared to Fox and Tanaka, this assumption is almost certainly violated. Further, as the formulas necessarily do not result in the same HRmax predictions, one can exclude the null hypothesis a priori, which is why performing the ANOVA makes no sense. The main effect is necessarily ≠ 0.
I would rather suggest comparing the differences between predicted and actual HRmax between formulas, as shown in the right part of the figure. Importantly, a test comparing central tendencies (as means or medians) will not suffice, as shown by the following example: Assume that the mean (!) difference from the actual HRmax is exactly 0 in both formulas. This would necessarily result in a n.s. test comparing mean deviations between formulas. However, even under this assumption the dispersion around 0 could be highly different between formulas, making the one with the smaller dispersion much better in predicting HRmax. Thus, I would recommend a statistical analysis that tests whether the deviation from the real HRmax is systematically lower on one of the two formulas. For example, one could square each data point’s difference from 0 and test with a paired test if one formula results in lower squared deviations. I would definitely recommend consulting a statistician for this question. Bland-Altmann Plot: The plot does not make sense in its current form. It is not clear why the x-axes are Averages – averages of what? I interpret that these are the actual HRmax values based on line 114 and 115, and it is circular reasoning that ‘there was an overestimation when actual HRmax was low and an underestimation when actual HRmax was high’. Instead I would suggest a single scatter plot with maturation on the horizontal x-axis (if used as continuous variable, see below), and the (squared) differences (Delta HR) on the y-axis. Then, I would plot data points resulting from both formulas in this diagram, with different colours or symbols for each formula. Next, one could test whether the deviation from the true HRmax values differs between groups after adjustment for maturation using an ANCOVA, but not a usual one, rather with a within-subjects factor ‘formula’. In SPSS, such an analysis can be done using mixed linear models. One could see this analysis as a paired t-test with a continuous covariate, whereas the “classical ANCOVA” rather represents an unpaired ttest with a continuous covariate. Again, I would recommend consulting a statistician for this purpose. It would be important to report the interaction term (which should be removed from the model if n.s.) and both main effects. The resulting regression lines (non-parallel in case of a sig. interaction, or parallel in case of no interaction) should be added to the above-mentioned diagram. Groups of less and more matured participants: It is unclear how these groups were defined. Is there a categorical variable that a priori defines groups? Or was an arbitrary cut-off chosen (which is unfortunately often done but should be avoided, see PMID: 16217841). I have the impression that with my above-mentioned suggestion regarding an ANCOVA with a within-subjects factor the question regarding the effect on maturation on validity is answered, however, if there is more to be analysed, this should – as well – be discussed with the statistician. Figure 1: Can be omitted, a correlation would seem to unlikely and is dispensable for the research question. The discussion should be re-written after the new analyses, with a focus on which, if any, formula is better in the assessed population. It is not clear what the primary aim was: Was it to assess whether maturity affects validity? Then, this should be included in the title. Otherwise, maturation should not be mentioned so prominently at the end of the introduction, where the main aim / hypothesis should be clearly recognizable. It should be described in more detail how peak height velocity was used. It should be clear without consulting the paper by Mirwald et al. In addition, Mirwald et al state in their conclusion that “It is recommended that maturity offset be considered as a categorical rather than a continuous assessment.” I do not claim that this recommendation must be followed, however, a reasonable justification should be provided for the decision. Line 60: It is not clear what ‘selected to be considered as members of national teams’ means Line 136: “…it was assumed that the older participants would exhibit lower HRmax than their younger counterparts due to their larger accumulated training experience.” Does training reduce the maximal heart rate? If yes, provide evidence from a randomized controlled trial (not an observation study). If not, this point should be removed. “…indicating that volleyball could not induce chronic adaptations similar as endurance training.” I disagree. A randomized controlled trial would have been necessary to test this hypothesis. Line 144: non-statistically significant: Probably you mean statistically non-significant.

Reviewer 2 Report

Overall review

A solid paper with a clear research question. The data reported for the Bland-Altman analysis was excellent, as most agreement studies omit confidence intervals and SD of the differences. 

A few references were omitted that also addressed the effect of maturation on HRmax in children and adolescents.

For the statistical methods, I believe the inclusion of an ANCOVA may not be appropriate here, since the covariate (age) does not appear to predict the outcome variable (HRmax). Additionally, the assumptions of the ANCOVA are not discussed in the methods section.

Even without the ANCOVA, the inclusion of maturation makes this paper novel (although not entirely unique). The meta-analysis cited in this paper discusses the potential importance of accurately stratifying under-18 subjects by pubertal status, so it is good to see researchers working towards this suggestion.

Introduction

Lines 51-54: This idea about the role of maturation should be expanded to better set up the purpose. I am aware of at least two papers which used the effect of maturity in trying to determine HRmax in adolescents; including them here (and in the discussion) would help demonstrate the importance of including the role of maturation in HRmax. Consider the following papers, please:

Gelbart M, Ziv-Baran T, Williams CA, Yarom Y, Dubnov-Raz G. Prediction of maximal heart rate in children and adolescents. Clinical Journal of Sport Medicine. 2017;27(2):139-44.

Grouped participants by pubertal status based on age

Mahon AD, Lee JD, Hanna LE. Evaluating the prediction of maximal heart rate in children and adolescents. Research Quarterly for Exercise and Sport. 2010;81(4):466-71.

Determined maturity offset and included it in regression analysis

Methods

Lines 88-89: Missing some inequality/equal signs for very small and nearly perfect correlations.

Lines 91-93: I'd recommend including the assumptions required for ANCOVA (there are a lot!) to assure the reader that you've done your due diligence. 

Further about the ANCOVA: In order to perform an ANCOVA, the covariate (age in this case) needs to be a significant predictor of the outcome variable; if age is not a significant predictor of HRmax (as suggested in Figure 1, but a regression predicting HRmax from age would be more appropriate to test this), then it is not an appropriate covariate. I may be misunderstanding the methods outlined here, but from what I can tell the ANCOVA may not be entirely justified.

Results

Lines 101-102: Not sure I would consider p=0.104 to be close to statistically significant. This may be a preference thing, but I would suggest removing the "close to" and just report the p-value for the ANCOVA. 

Table 2: Very happy to see researchers reporting mean and SD of the differences in these sorts of papers!

Lines 111-112: Also glad to see 95% CI reported around the mean differences. Most researchers omit these, even though Bland and Altman specifically stressed the importance of them in their original paper. 

Figure 2: The presentation of the data in these graphs is very good. Maybe just explain a bit more in the figure caption what's going on (e.g., the error bars, the individual points, etc.).

Discussion

Lines 126-128: I think you should expand upon the limits of agreement for both equations - they are around ±16 bpm for each equation! That is a large amount of individual variation. Even the Tanaka equation, with a smaller mean difference, has a +30 beat range in which the differences for 95% of the sample were observed. The limits of agreement should also be compared to other research done in this area. The purpose of the Bland-Altman analysis isn't just to show mean differences, it is to show the range of differences that can be expected between two methods. Discussing this will strengthen your overall conclusions. 

Line 133: "Furthermore, HR has been under the control of autonomous nervous system[...]"

Weird sentence structure. Heart rate is controlled by the autonomic nervous system. Maybe delete the first part of the sentence and leave it as, "Furthermore, growth related differences in HR might be linked to...".

Lines 145-146: Again, careful with the statement, "close to statistical significance". Focus more on the effect size, which is more informative than a p-value anyway. A power analysis may be useful here; was your sample size large enough to detect a significant effect between groups? If the sample is under-powered, then that may be why a medium effect size was observed in spite of a non-significant result. I would recommend adding in a power analysis post-hoc to calculate the power achieved by your study design. This will also allow you to comment that despite being non-significant, a moderate effect was detected (which is arguably more important anyway).

Lines 144-149: Going back to the studies I recommended earlier, this would be a good place to compare your results with theirs. Both studies discuss the effects of maturation (though defined differently) on HRmax in children and adolescents. 

Summary of recommendations

Better expand on the idea of maturation affecting HRmax by including the referenced studies that also examine it. Double check the ANCOVA assumptions - age cannot be included as a covariate if it is not significantly predictive of HRmax. Include a power analysis to justify your sample size - was it large enough to detect the effects you were looking for? Discuss the actual limits of agreement in the discussion (approximately ±16 bpm).  Compare your results with other studies in this area to further support your conclusion.
